# Controlling the Thermal Conductivity of Monolayer Graphene with Kirigami Structure

**DOI:** 10.3390/membranes12111128

**Published:** 2022-11-10

**Authors:** Yuan Gao, Shuaijie Lu, Weiqiang Chen, Jinyuan Zhang, Chundi Feng, Yanming Liu

**Affiliations:** 1School of Transportation and Civil Engineering, Nantong University, Nantong 226019, China; 2Department of Mechanical, Aerospace and Civil Engineering, School of Engineering, The University of Manchester, Manchester M13 9PL, UK; 3School of Life Sciences, Nantong University, Nantong 226019, China; 4School of Public Health and Preventive Medicine, Monash University, Melbourne, VIC 3004, Australia

**Keywords:** graphene-based membrane, kirigami structure, molecular dynamics simulation, thermal conductivity, adjustability

## Abstract

In this work, the thermal conductivity performance of graphene kirigami (GK) was systematically investigated via molecular dynamics (MD) simulations. The results indicate that the degree of defects (DD) on GK has a significant influence on thermal conductivity. Reducing the DD is the most effective way to decrease the thermal conductivity of GK. For zigzag-incised GK sheets, the change rate of thermal conductivity can reach up to 1.86 W/mK per 1% change in DD by tuning the incision length. The rate of changing thermal conductivity with DD can be slowed down by changing the width among incisions. Compared with the zigzag-incised GK sheets, heat transfer across the armchair-incised GK comes out more evenly, without significant steep and gentle stages along the heat transfer routes. More importantly, the GK structure can adjust the thermal conductivity by stretching, which the previously reported nanoporous graphene does not have. The change rate of thermal conductivity achieves about 0.17 W/mK with 1% stretching strain for simulated GK and can be further reduced at high tensile strain rates, benefiting the precise and variable control of the thermal conductivity of the monolayer graphene.

## 1. Introduction

Graphene, as a new two-dimensional nanomaterial, has exhibited superior electrical, mechanical, optical, and electrochemical properties [1,2,3]. Due to high carrier mobility, excellent thermal conductivity, and ultra-high Young’s modulus, graphene-based membranes have excellent application prospects in heat transfer [4], optoelectronics [5], sensors [6], and energy storage [7]. Previous studies have suggested that the thermal performance of graphene sheets can be adjusted via several methods, such as tuning edges [8], pores [9], and boundary conditions [10]. Changing the pore size distributions (including pore size and degree of defects) is the most common and significant approach to adjusting the thermal properties of graphene-based membranes in previous studies [11,12]. Regulating the thermal conductivity ability of graphene-based membranes by perforation inspires potential applications in thermal management and thermoelectric nanodevices [13,14].

In the last decades, some scholars have investigated the defect-dependent thermal properties of nanoporous graphene (NPG) membranes [15]. Zhang et al. [16] employed non-equilibrium molecular dynamics simulations to study the thermal conductivity of defective graphene and revealed that Stone–Wales defects could significantly drop the thermal conductivity of graphene sheets. The decreasing rate highly depended on the density and types of defects. Mortazavi and Ahzi [17] further researched the influence of point vacancy, Stone–Wales and bivacancy defects on monolayer NPG membranes’ thermal properties using MD simulation. They indicated that the thermal performance of the NPG is sensitive to defect distribution. Malekpour et al. [18] reported the thermal conductivity of suspended graphene as a function of defects density via the Boltzmann transport equation and MD simulations. However, although several studies have proved that perforating monolayer graphene as an NPG can adjust its thermal conductivity, the adjustment is fixed and irreversible [19]. Furthermore, finely controlling the NPG membrane’s pore size is highly cost- and time-consuming [2,15,20]. Therefore, more efficient methods to tune the thermal conductivity of the graphene-based membrane should be put forward.

Here, we propose to use graphene kirigami (GK) as an adjustable thermal conductivity membrane. Kirigami, derived from the traditional art of origami, can transform two-dimensional structures into three-dimensional structures [21,22]. In recent years, kirigami has been applied to advanced materials and has changed the mechanical, thermal, and electrical properties of these materials [23,24]. GK is a variety of NPG membranes that retains the superior properties of graphene-based membranes, including single-atom thickness and honeycomb structure [25,26]. Additionally, GK endows traditional NPG membranes with properties they do not have, such as ultra-high flexural performance, stretchable capability, and adjustable pore sizes [27]. However, GK membranes have not been widely used in practical projects mainly due to the limitation of their fabrication technologies [27]. Previous research reported that some cutting-edge methods could fabricate GK membranes, including optical lithography [25], focusing on beam [28] and irradiation-induced technologies [29]. Nevertheless, due to the issues of manufacturing speed, cost, and accuracy, GK membranes still have difficulty in full-scale production [27]. Fortunately, in the authors’ recent study, we found a cost-efficient and time-saving method for the large-scale preparation of GK membranes called selective tearing [27]. Furthermore, the selective tearing generated by GK membranes has been proven efficient in gas separation [27] and water desalination [30]. Considering the superior properties of GK membranes, they have great potential to be applied in electrochemical capacitors, sensors, battery technologies, energy harvesters, electrochemical supercapacitors, and micromotors.

Hence, in this work, we provide a systematical investigation of the thermal properties of GK, particularly the relationship between the deformation of GK and its corresponding thermal conductivity. First, a model of monolayer GK with nanoscale incisions was established according to the literature [31]. After that, the thermal behavior of GK was studied using large-scale atomic molecular massively parallel simulator (LAMMPS) software, manufactured by Sandia National Labs, Albuquerque, NM, USA [11]. The effects of incision size, incisions distribution, degree of defects (DD), conductive direction, and stretching rate on the heat flux, temperature gradient, and thermal conductivity of GK are reported. The approaches concerning precisely controlling the thermal conductivity of monolayer graphene-based membranes through the kirigami structure are discussed.

## 2. Methods

### 2.1. Model Establishment

The setup of the hydrogenated GK with both zigzag and armchair incisions is illustrated in Figure 1. During the model construction process, two horizontal columns of carbon atoms in the dotted pane on the graphene sheet with a periodical boundary were removed to create incisions, as exhibited in Figure 1a. Afterwards, the hydrogen atoms were supplemented at the broken bonds, establishing the GK unit. After that, the GK unit was supercell with the size of 51.12 × 49.2 Å^2^ with the periodical boundary. The models of the generated GK with zigzag and armchair incisions are presented in Figure 1b,c, respectively. The DD of the GK can be tuned by two parameters named the incision length (L) and the width among incisions (W), as shown in Figure 1b,c. The simulated GK with different incision lengths and the width among incisions in both zigzag (ZZ) and armchair (AC) incisions are listed in Table 1. The DD of the GK is defined according to previous literature [31,32], equaling the ratio of the removed carbon atoms to the total atoms on the entire graphene sheet.

### 2.2. Simulation Details

Following the methodology discussed in previous studies [11,33,34,35,36], we adopted nonequilibrium molecular dynamics (NEMD) simulations to study the thermal response of the built GK. As is shown in Figure 2, two regions with the width of 0.5 nm and centered at $x=L_{x}/4$ and $x=3L_{x}/4$ were placed in Langevin thermostats [37] with a high temperature ($T_{H}=350\;K$) and a low temperature ($T_{L}=250\;K$), which resulted in a temperature gradient ∇T and heat flux q along the x direction. The heat flux q was calculated by following a previous study [34,35]. The thermal conductivity k was thus calculated from Fourier’s law for heat conduction k=−q/∇T. The reactive empirical bond order (REBO) potential [38] was used to describe the interatomic interactions in GK with a timestep of 1 fs, which has proved to accurately predict the thermal response of graphene-based membranes (e.g., [39,40]). Periodic boundary conditions were applied in all directions. The NEMD simulation was performed by following the following procedure: first, the GK experiences an energy minimization using a steepest descent integrator. Second, a relaxation run for the whole system in NVT ensemble with the temperature of $(T_{H}+T_{L})/2=300\;K$ lasting 1 ns is performed. Third, the two predefined thermostats are switched on and an equilibrium run of 1 ns is performed to obtain the steady state, where the regions between the two thermostats are in NVE ensemble, and the two thermally-bathed GK ribbons are in NVT ensemble, i.e., an NVE ensemble together with a Langevin thermostat. Finally, a production run lasting 1 ns is performed to collect the thermodynamic data and temperature distribution profiles. The GK models are divided into 70 bins along the x directions to collect the data. Note that the measured thermal conductivity is under a vacuum environment. If the studied GK is immersed in the environment of solvent molecules, the nearby solvent molecules of the GK layers will act as an additional heat transfer media and fill in the defects of GK layers. Therefore, the thermal conductivity will be increased. However, if it is an aqueous environment, as the thermal conductivity of water is much lower, the increment will be trivial.

## 3. Results and Discussion

### 3.1. DD Effects

The effect of the DD on the thermal conductivity of the monolayer GK is significant. The previous study indicated that the heat transfer in the graphene-based membrane was sensitive to defects’ and voids’ distribution [17,41]. Hence, this section investigates the DD effects on the thermal conductivity of GK. The DD of the GK can be changed in two main ways: tuning the incision length and changing the width among incisions. Here, we marked the first influence concerning the incision length as Group_1 and the second effect as Group_2 to distinguish them better. The corresponding thermal performance of the two groups was then studied. As presented in Figure 3a, the heat flux increases linearly with the simulation time, illustrating a uniform heat transfer on GK. The time-averaged temperature distribution of each atom along the direction perpendicular to GK incisions of Sample ZZ_1_1 during the entire simulation process is demonstrated in Figure 3b. The temperature gradient from the hot zone to the cold zone also showed a trend of linear decrease, with a goodness coefficient of 0.97. This phenomenon reveals that the thermal conductivity is uniform overall using GK as the heat conduction material. Additionally, for the data of the fitting line, the steep stage and gentle stage take turns to appear. Combined with the snapshots of the temperature distribution of Sample ZZ_1_1 after one nanosecond heat conduction exhibited in Figure 3c, it can be concluded that the thermal conductivity of the unbroken graphene area is high. By contrast, the thermal conductivity on the area of the incisions is low. This finding highly agrees with previous reports [12], proving that the thermal conductivity of the graphene-based membrane can be reduced by creating incisions on the GK.

The detailed heat transfer process on the zigzag-incised GK is shown in Figure 3c. The GK conducts heat via the carbon atoms connected at both ends of the incisions. The carbon atoms in this area are relatively uniform in temperature. On the contrary, the temperature distributions of the unbroken graphene area are nonuniform. As exhibited in Figure 3c, from 2.1 to 3 nm on the X-axis, the temperature of the carbon atoms fluctuates along the Y-axis, demonstrating the apparent temperature transfer gradient. Additionally, the carbon atoms at the edge of a single incision have a similar temperature, suggesting that these carbon atoms are not on the main path of heat transfer and contribute less to thermal conductance. Figure 3d further shows the relationship between the DD and the thermal conductivity on the zigzag-incised GK of Group_1. As the DD increases, the thermal conductivity of GK decreases linearly, with a high fitting coefficient (*R*^2^ = 0.985). This is mainly due to the fact that heat transfer in the graphene sheet results in lattice vibration by phonons, which is sensitive to the defects and voids on the graphene [41]. Pores on the graphene sheet effectively decrease phonon transport [11]. Thereby, the thermal conductivity of GK reduces with decreasing DD. When the thermal performance of the zigzag-incised GK is changed through the incision length, the change rate of thermal conductivity with DD is about 1.86 W/mK.

Adjusting the width among incisions is another efficient approach to tuning the thermal conductivity of GK. Figure 4 presents the time-averaged temperature distribution of each atom along the direction perpendicular to Sample ZZ_2_1, Sample ZZ_2_2, and Sample ZZ_2_3, and their corresponding temperature distribution after one nanosecond of heat conduction simulation. It can be found that with the increasing distance between the incisions, the linearity of the thermal conduction fitting is gradually dropped. The steep and gentle stages become apparent in Sample ZZ_2_3, indicating that as the distance between the incisions rises, the uniformity of the thermal conductivity on the monolayer GK worsens. The results of the snapshots of the temperature distribution agree with this finding. For Sample ZZ_2_1, the heat transfer between the carbon atoms is much more uniform. On the contrary, for sample ZZ_2_3, since some carbon atoms are not on the main thermal conduction route, they show significant temperature differences from their neighbors.

Figure 5 demonstrates the relationship between the DD and the thermal conductivity on the zigzag-incised GK. For Group_2, a quadratic function exists between the thermal conductivity and DD, with a goodness coefficient *R*^2^ = 0.999. The change rate of thermal conductivity of the zigzag-incised GK in Group_2 is about 0.75 W/mK between 10% and 20% DD. Compared with the change rate of thermal conductivity in Group_1, the value reduces by about 59.7%. The change rate of thermal conductivity would be lower than 0.75 W/mK at high DD. This phenomenon indicates that controlling the thermal conductivity of monolayer GK by changing the width among incisions is more accurate. However, changing the incision length provides greater amplitude regulation to adjust the thermal conductivity of GK. Figure 5b presents the relationship between the DD and the thermal conductivity of all established GK membranes. There is a quadratic correlation among these membranes with a coefficient value of 0.961. Compared with Figure 3d and Figure 5a, the fitting coefficient decreases slightly, suggesting that the thermal conductivity of the zigzag-incised GK is not only related to the DD but also to the incision distributions.

By exchanging the positions of the two thermostats, we calculated the thermal conductivity of the studied GKs towards the negative x direction and the results are listed in Table 2. No thermal rectifications are observed and the thermal rectification factors are nearly zero, that is, the thermal conductivity from the left to the right is the same as that from the right to the left. In addition, the computed thermal conductivity of the GK layers is in the same order of magnitude to that of nano-porous graphene layers with a similar porosity, measured by a previous study [11], validating its correctness.

### 3.2. Heat Transfer Direction Effects

Due to the unique chirality, graphene has different thermal and electrical conductivity characteristics in the zigzag and armchair directions [42]. This section discusses the influence of the incision directions on thermal conductivity. The time-averaged temperature distribution of atoms along the direction perpendicular to the incisions of armchair-incised GK Sample AC_3_1 is shown in Figure 6a. Compared with the zigzag-incised GK, heat transfer through the armchair-incised GK performed more evenly. There is no apparent difference between steep and gentle stages. The linear fitting coefficient of the temperature gradient of Sample AC_3_1 hits the value of 0.99, approximately 2% higher than that of Sample ZZ_2_3 with similar DD. The relationships between the DD and the thermal conductivity on the armchair-incised GK in Group_3 are shown in Figure 6c. A linear correlation with a fitting coefficient of 0.938 was obtained. The change rate of thermal conductivity of Group_3 is about 1.54 W/mK, whose value is between Group_1 and Group_2. Figure 6d lists the DD and thermal conductivity of all of the simulated GK membranes without stretching. It can be found that under the same DD, the zigzag-incised GK always has higher thermal conductivity than the armchair-incised GK. Fitting the thermal conductivity and DD of all simulated GK membranes, a weak quadratic correlation with only a 0.663 fitting coefficient was obtained. The low fitting coefficient indicates that different incision directions significantly influence the thermal conductivity of GK.

### 3.3. Tensile Strain Effects

By virtue of superior adjustability and the advantages of graphene-based nanosheets as monolayer membranes, the GK structure can tune its thermal conductivity through stretching, which the previously reported NPG cannot [43]. Figure 7 presents the time-averaged temperature distributions and corresponding snapshots of the temperature color maps along the direction perpendicular to the zigzag-incised GK incisions under 5%, 15%, and 25% strain. With stretching, the heat transfer fluctuates more than the unstretched state under low strain. The increasing strain makes the difference between steep and gentle stages more obvious. The shape of the temperature profile in Figure 7a is determined by the structural characteristics of GK layers after stretching. The stretching process will lead to the generations and disappearances of wrinkles on GK layers [27], which will affect the heat transport path of the GK layers. The smaller fitting coefficient $R^{2}$ value indicates the less-uniform distributions of pores and deformations. Therefore, the temperature gradient fitting coefficient of Sample ZZ_1_1_15% is lower than the other two in Figure 7a because of its less-uniform distributions of pores and deformations at this strain value. The color map of temperature distribution proves this view. As shown for Sample ZZ_1_1_5% in Figure 7b, the temperature difference between some carbon atoms and the surrounding carbon atoms is significant, indicating that these carbon atoms are not on the primary heat conduction path of the GK and contribute little to thermal conduction. Nevertheless, with the increase of tensile strain, the GK produces greater deformations and generates wrinkles to guide thermal conduction. Therefore, the thermal conductivity becomes uniform for Sample ZZ_1_1_25%.

The thermal conductivity of the stretched armchair-incised GK demonstrates similar characteristics. As presented in Figure 8, the fitting coefficient of the temperature gradient tends to drop first and then grow up. The difference between the steep and gentle stages is most significant at 15% stretching strain. According to previous literature [27], the ultimate tensile strain of the zigzag-incised and armchair-incised GK could reach 40% and 35% respectively. Hence, in this work, we employed Sample ZZ_1_1_25% and Sample AC_3_1_15% for thermal conductivity simulation to ensure that the C–C bonds on GK do not fracture and keep stable during heat transfer.

The relationship between the strain rate and the thermal conductivity on both zigzag-incised and armchair-incised GK was calculated and is shown in Figure 9. Two different fittings occur on the GK with different incision directions. A high quadratic function fitting with a 0.994 fitting coefficient was obtained for Sample ZZ_1_1 with the stretching strain between 5% to 25%. The change rate of thermal conductivity is about 0.17 W/mK with 1% stretching strain. The change rate of thermal conductivity can be further reduced at high tensile strain rates, achieving approximately 0.076 W/mK with 1% stretching strain between 20% and 25%. For the armchair-incised GK (Sample AC_3_1), the change rate of thermal conductivity is about 0.18 W/mK with 1% stretching strain, which is slightly lower than that of the zigzag-incised GK. However, the thermal conductivity drops linearly with the strain, meaning it is more controllable. Overall, the fitting results in shown Figure 9 prove that the thermal conductivity can be precisely regulated by strain.

Overall, this study’s results propose that the thermal conductivity of the graphene can be adjusted with the kirigami structure. Two main findings were obtained in this work. On the one hand, the DD significantly affects the heat transfer efficiency of GK. This finding highly agrees with previous studies [16,34,40]. Fang et al. [11] reported that compared to the intact graphene sheet, the thermal conductivity of the NPG dropped by approximately 90% under 25% DD. Yousefi et al. [12] revealed that heat transfer directions also greatly influence the thermal conductivity of NPG sheets. Similar simulation results can be found in our study. For Sample ZZ_2_1, the thermal conductivity only equals 3.3 W/mK under 20% DD. By contrast, for Sample ZZ_1_4, the thermal conductivity increases to 17.6 W/mK with 6% DD, increasing by about 533%. On the other hand, GK could tune the thermal conductivity by stretching under the same DD, which the previously reported NPG cannot. This property not only allows GK to adjust its thermal conductivity without changing the structure but also can precisely adjust the thermal conductivity of the material.

## 4. Conclusions

In this study, the thermal conductivity of GK was investigated via MD simulations. The results indicate that the effects of DD, incision directions, and stretching strain on the thermal properties of the monolayer GK are significant. Tuning the DD in different ways, including changing the incision length and width among incisions, also influences the thermal conductivity of GK. The thermal conductivity of zigzag-incised GK exhibits a linearly decreasing trend with the reduction of the incision length, with a fluctuation between 7.1 and 16.7 W/mk under DD ranging from 6.07% to 11.59%. Tuning the thermal conductivity of GK by changing the width among incisions is another effective method. With the increasing distance between the incisions, the linearity of the thermal conduction fitting is gradually dropped according to a quadratic function with a goodness coefficient *R*^2^ = 0.999. The change rate of thermal conductivity under this condition is about 0.75 W/mK between 10% and 20% DD. Compared with the change rate of thermal conductivity of zigzag-incised GK adjusted by incision length, the change rate is reduced by about 59.7%. The simulation results of Group_1 and Group 2 suggest that controlling the thermal conductivity of monolayer GK by changing the width among incisions is more accurate. Nevertheless, changing the incision length provides greater amplitude regulation to adjust the thermal conductivity of GK. Compared with the zigzag-incised GK, heat transfer across the armchair-incised GK comes out more evenly, without significant steep and gentle stages along the heat transfer routes. Under the same DD, the zigzag-incised GK always has higher thermal conductivity than the armchair-incised GK. By its superior adjustability, GK structure can adjust the thermal conductivity by stretching, which the previously reported NPG cannot. A quadratic function fitting and a linear fitting were obtained for zigzag-incised and armchair-incised GK, respectively. The change rate of thermal conductivity is about 0.17 W/mK with 1% stretching strain for Sample ZZ_1_1. The change rate of thermal conductivity can be further reduced at high tensile strain values, which is very conducive to the accurate control of the thermal conductivity of GK. For the armchair-incised GK, the change rate of thermal conductivity is about 0.18 W/mK with 1% stretching strain. Although a little bit lower than that of the zigzag-incised GK, the thermal conductivity shows a linear relationship and decreases steadily with the stretching rate.

Overall, this work demonstrated the thermal conductivity of GK. It could inspire the potential application of GK membranes in a broad spectrum of fields, including electrochemical capacitors, sensors, battery technologies, energy harvesters, electrochemical supercapacitors, and micromotors.

## Figures and Tables

**Figure 1 membranes-12-01128-f001:**
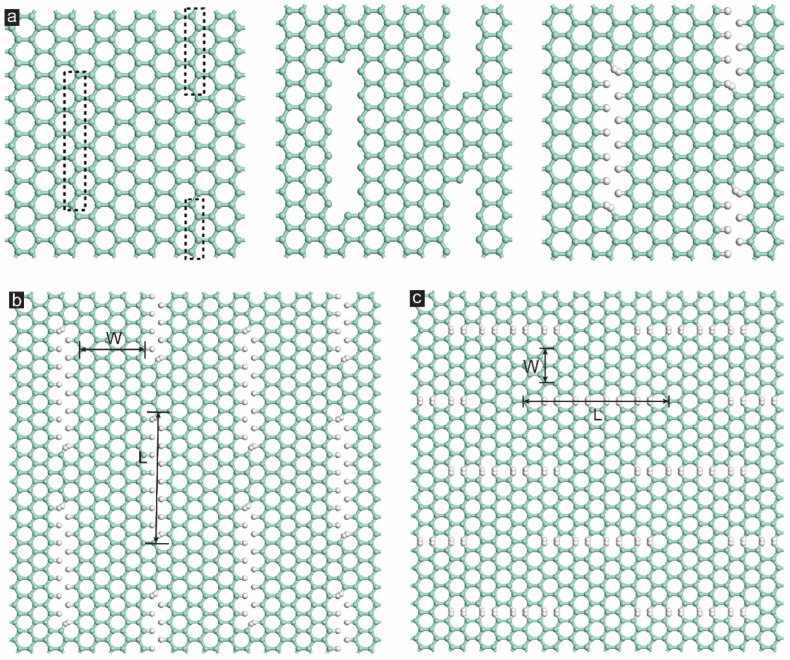
(**a**) Creating a zigzag-incised graphene kirigami (GK). The carbon atoms in the dotted line were deleted and all bonds were capped using hydrogen. The monolayer GK with (**b**) zigzag incisions and (**c**) armchair incisions [29].

**Figure 2 membranes-12-01128-f002:**
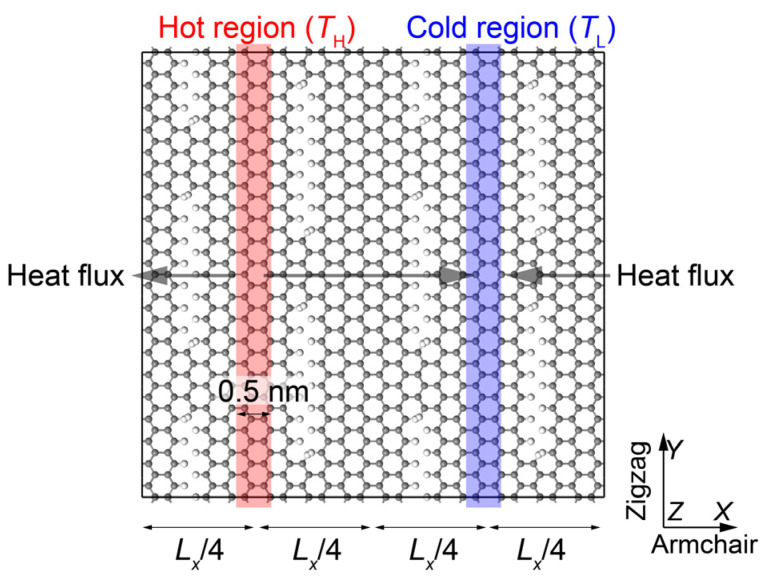
The setup for nonequilibrium molecular dynamics (NEMD) simulations to measure the thermal conductivity of GK.

**Figure 3 membranes-12-01128-f003:**
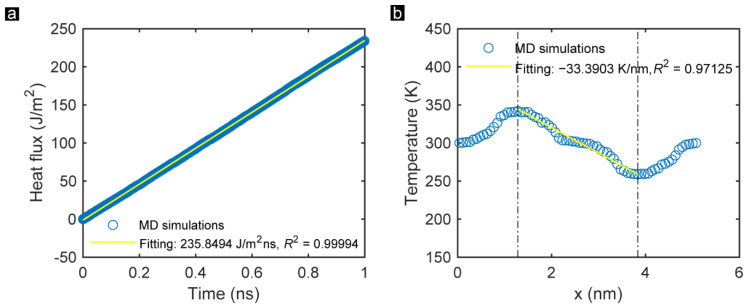
(**a**) Fitting results of the heat flux on the monolayer GK of Sample ZZ_1_1 within one nanosecond. (**b**) Time-averaged temperature distribution of each atom along the direction perpendicular to the GK incisions during the simulation. (**c**) Snapshots of the temperature distribution of Sample ZZ_1_1 after one nanosecond of heat conduction. (**d**) Relationship between the DD and the thermal conductivity on the zigzag-incised GK in Group_1.

**Figure 4 membranes-12-01128-f004:**
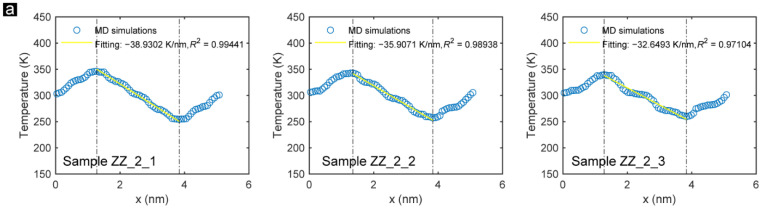
(**a**) Time-averaged temperature distribution of each atom along the direction perpendicular to the zigzag-incised GK incisions (Sample ZZ_2_1, Sample ZZ_2_2, and Sample ZZ_2_3) during the simulation process. (**b**) Snapshots of the temperature distributions of Sample ZZ_2_1, Sample ZZ_2_2, and Sample ZZ_2_3 after one nanosecond of heat conduction.

**Figure 5 membranes-12-01128-f005:**
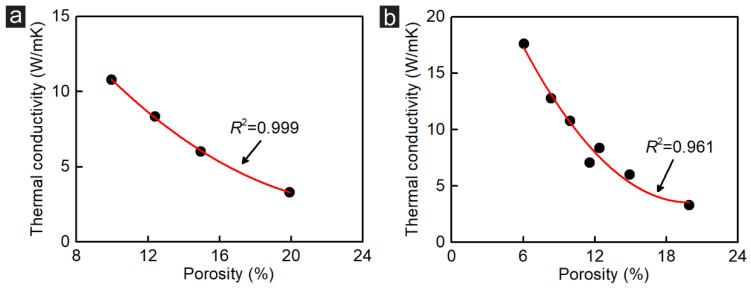
Relationship between the DD and the thermal conductivity on zigzag-incised GK (**a**) in Group_2 and (**b**) in both Group_1 and Group 2.

**Figure 6 membranes-12-01128-f006:**
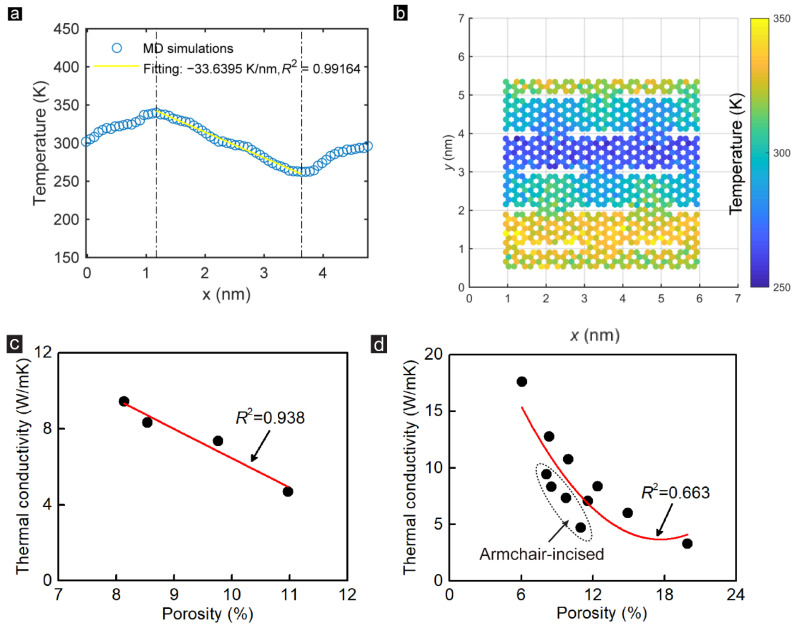
(**a**) Time-averaged temperature distribution of each atom along the direction perpendicular to the armchair-incised GK incisions (Sample AC_3_1) during the simulation process. (**b**) Snapshots of the temperature distribution of Sample AC_3_1 after one nanosecond of heat conduction. Relationship between the DD and the thermal conductivity on (**c**) armchair-incised GK in Group_3 and (**d**) all established GK membranes without stretching.

**Figure 7 membranes-12-01128-f007:**
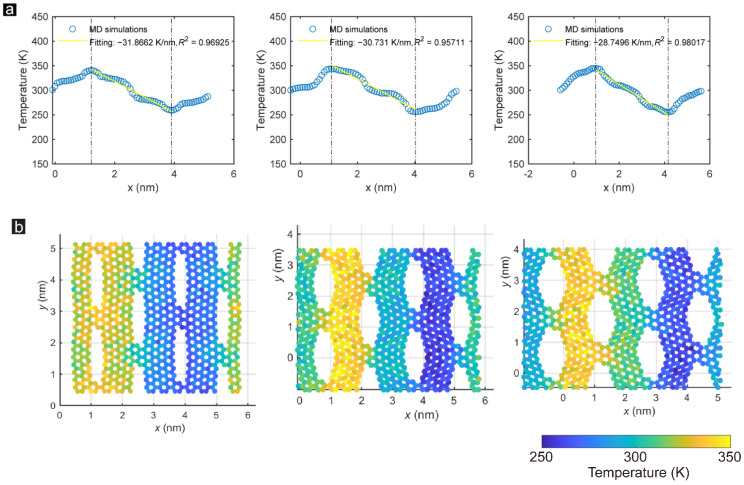
(**a**) Time-averaged temperature distribution of each atom along the direction perpendicular to the zigzag-incised GK incisions under 5%, 15%, and 25% strain. (**b**) Snapshots of the temperature distribution of Samples ZZ_1_1_5%, ZZ_1_1_15%, and ZZ_1_1_25% after one nanosecond of heat conduction.

**Figure 8 membranes-12-01128-f008:**
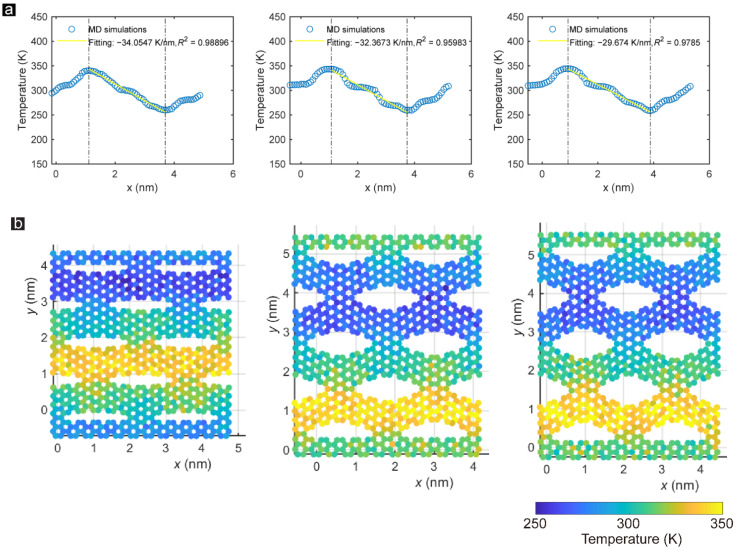
(**a**) Time-averaged temperature distribution of each atom along the direction perpendicular to the armchair-incised GK incisions under 5%, 15%, and 20% strain. (**b**) Snapshots of the temperature distribution of Samples AC_3_1_5%, AC_3_1_15%, and AC_3_1_20% after one nanosecond of heat conduction.

**Figure 9 membranes-12-01128-f009:**
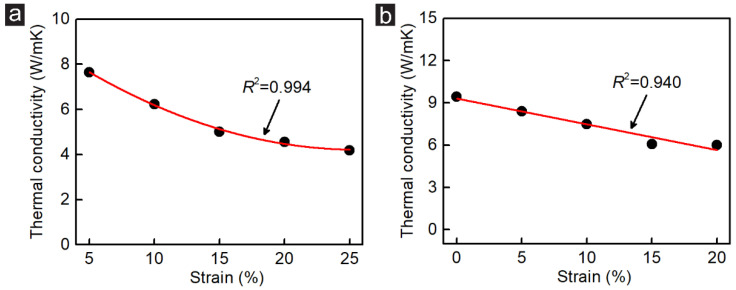
Relationship between the strain rate and the thermal conductivity on the (**a**) zigzag-incised GK (Sample ZZ_1_1) and (**b**) armchair-incised GK (Sample AC_3_1).

**Table 1 membranes-12-01128-t001:** Design parameters of the monolayer GK sheets.

Sample	L (Å)	W (Å)	DD (%)
ZZ_1_1	18.46	8.52	11.59
ZZ_1_2	16.01	8.52	9.96
ZZ_1_3	13.55	8.52	8.33
ZZ_1_4	11.09	8.52	6.07
ZZ_2_1	16.01	2.84	19.92
ZZ_2_2	16.01	4.26	14.94
ZZ_2_3	16.01	7.10	12.4
AC_3_1	19.88	4.92	8.13
AC_3_2	17.04	2.46	8.54
AC_3_3	19.88	2.46	9.76
AC_3_4	21.30	2.46	10.97

**Table 2 membranes-12-01128-t002:** Measured thermal conductivity of the monolayer GK. The values in parentheses were computed by exchanging the positions of two thermostats.

Sample	Heat Flux (J/m^2^·ns)	Temperature Gradient (K/nm)	Thermal Conductivity (W/mK)
ZZ_1_1	235.8 (245.2)	33.4 (34.0)	7.1 (7.2)
ZZ_1_2	327.1 (322.6)	30.3 (30.4)	10.8 (10.6)
ZZ_1_3	357.0 (356.6)	27.9 (27.8)	12.8 (12.8)
ZZ_1_4	435.2 (439.1)	24.7 (24.7)	17.6 (17.8)
ZZ_2_1	128.7 (128.4)	38.9 (39.3)	3.3 (3.3)
ZZ_2_2	216.0 (222.4)	35.9 (36.1)	6.0 (6.2)
ZZ_2_3	272.7 (264.8)	32.6 (31.9)	8.4 (8.3)
AC_3_1	317.0 (314.5)	33.6 (33.7)	9.4 (9.3)
AC_3_2	287.4 (283.1)	34.5 (34.7)	8.3 (8.2)
AC_3_3	264.5 (265.9)	35.9 (36.1)	7.4 (7.4)
AC_3_4	184.6 (177.6)	39.3 (38.2)	4.7 (4.6)
ZZ_1_1_5%	243.9 (235.8)	31.9 (31.4)	7.7 (7.5)
ZZ_1_1_10%	193.3 (199.8)	31.0 (32.9)	6.2 (6.1)
ZZ_1_1_15%	154.2 (153.0)	30.7 (30.4)	5.0 (5.0)
ZZ_1_1_20%	142.0 (135.9)	31.1 (29.0)	4.6 (4.7)
ZZ_1_1_25%	120.2 (117.2)	28.7 (28.4)	4.2 (4.1)
AC_3_1_5%	285.7 (268.4)	34.1 (30.9)	8.4 (8.7)
AC_3_1_10%	240.6 (223.4)	32.4 (29.7)	7.5 (7.5)
AC_3_1_15%	196.3 (188.9)	32.4 (32.0)	6.1 (5.9)
AC_3_1_20%	178.4 (184.9)	29.7 (31.8)	6.0 (5.8)

## Data Availability

The data presented in this study are available on request from the corresponding author.

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
