# Peer review of "Controlling the Thermal Conductivity of Monolayer Graphene with Kirigami Structure"

_membranes, 2022, doi:10.3390/membranes12111128_

Round 1
Reviewer 1 Report
The authors simulated the heat transfer capability of a monolayer graphene kirigami membrane by MD simulation. Changing the thermal conductivity of the GK membranes by adjusting the length and width of the incisions has been systematically investigated. In addition, it is innovatively proposed that strain can tune the thermal conductivity of GK membranes.
In general, this article has certain guiding significance for investigating thermal conductivity of the GK membranes, but the details of the article need to be revised. As your reviewer, I suggest that your manuscript needs minor revisions.
My specific comments/questions are as follows:
(1) In the page 5, “As the porosity decreased, the thermal conductivity of the GK membrane decreased linearly, with a high fitting coefficient”. Please explain why thermal conductivity decreases with decreasing porosity.
(2) Why is the length of the GK membranes strained at 15% lower than the other two in Fig. 7b? Why the fitting result at 15% has the smallest R2 value compared to 5% and 25%, please explain reason.
(3) In the page7. “This is mainly due to the shortening of the interval between the neighboring incisions due to the stretching deformation and the narrowing of the graphene area that can transfer heat”. Why does stretching shorten the interval between neighboring incisions, shouldn't it increase?
(4) In Table 1, the author just compared the case where the length and width are the same. It is recommended to supplement the ZZ and AC samples with the same porosity.
(5) I recommend one reference paper “Bifurcation Analysis of Periodic Kirigami Structure with Out-Plane Deformation, Journal of the Society of Materials Science, Japan, 2018 Volume 67 Issue 2 Pages 202-207”, although it is written in Japanese.
Reviewer 2 Report
The authors introduce a set of defects on graphene by deleting stripes of carbon atoms and inserting hydrogens to cap the dangling bonds created by the deletions. They name the resulted nanosheets as graphene kirigami (GK) membranes. They provide models of GK membranes with different degree of defects by changing the length of the deleted stripes, and the distance between the successive stripes. Next, the authors set two regions on the nanosheet with different temperatures and employ NEMD simulations to realize the heat flux between these regions. They show that the thermal conductivity is reduced by increasing the degree of the defects. They also apply a stress force to strain the structures and show that the conductivity reduces with the amount of the strain. The discussion is limited to the comparison of the temperature contour plots of the defected graphenes, and the correlation between the thermal conductivity and the degree of defects (%), or the strain imposed on the structures.
The authors refer to the graphene kirigami (GK) layers as membranes. It’s more accurate to say that the kirigami layers are seen as potential structural elements of composite membrane systems. To configure a membrane system, the layers may intercalate in intermediate layered structures or be deposited on solid substrates like in https://doi.org/10.3390/app12073460. In this regard, the authors simulate free - standing graphene monolayers rather than membrane composite systems.
The kirigami layers do not interact with the environment as if they are in vacuum. If the structures are embedded in a solvent, the solvent molecules will increase or decrease the conductivity. The authors should give a brief comment on how the interactions of the nearby molecules should contribute to the heat flux through the defected layers.
Defects are not pores. The definition of porosity is misinterpreted with the void defect (or carbon rim). Graphenes are extensively modeled as pore walls representing idealized slit-pore geometries of porous carbons with variable size and surface chemistry. For example, see doi.org/10.1080/08927022.2015.1032275. I suggest that the term “porosity (%)” should be replaced with the term “degree of defects (%)” throughout the manuscript.
Reviewer 3 Report
In the present work, the authors investigated the thermal properties of kirigami graphenes using molecular dynamics simulations. they demonstrated that the porosity of the GK membrane has a significant influence on thermal conductivity. Reducing the porosity is the most effective way to decrease the thermal conductivity of the GK membrane. I found this paper interesting for the journal readers just after some minor revisions:
1- The abstract section is long. The authors should mention the main outcomes of their work in the abstract.
2- I believe that the authors should use the Nose-Hoover thermostat instead of Langevin because Langevin is a stochastic thermostat due to the presence of the random force acting on atoms while the Nose-Hoover thermostat is deterministic, and the same factor rescales all velocities to control the system's temperature in each simulation step.
3- Did the authors use the NVE ensemble for the thermal bath ribbons?
4- The simulation section was written poorly. Please use the following reference to cook this section.
https://iopscience.iop.org/article/10.1088/1361-6528/ac733e/meta
https://www.sciencedirect.com/science/article/pii/S1386947722002429
5- The quality of figure 2 is poor.
6- I suggest to the authors calculate thermal rectification for their Kirigami graphene.
7- The authors calculated the thermal properties of kirigami graphene, but in several parts of the manuscript they mentioned membrane. I didn't find that why they used the kirigami graphene membranes word. Please eliminate all the membrane words throughout the manuscript.
8- all the plots in figure 4 should be in the same size.
9- The main outcome of this research should be validated with the previous papers.
Round 2
Reviewer 2 Report
the manuscript is ready for publication in Membranes